# Adaptive Heritage: Is This Creative Thinking or Abandoning Our Values?

**Jim Perry** [1,*] and **Iain J. Gordon** [2]

1 Department of Fisheries, Wildlife and Conservation Biology, University of Minnesota, 2003 Upper Buford Circle, St. Paul, MN 55108, USA
2 Fenner School of Environment and Society, The Australian National University, Canberra, ACT 0200, Australia; iain.gordon@anu.edu.au
* Correspondence: Jperry@umn.edu

**Abstract:** Protected areas, such as natural World Heritage sites, RAMSAR wetlands and Biosphere Reserves, are ecosystems within landscapes. Each site meets certain criteria that allow it to qualify as a heritage or protected area. Both climate change and human influence (e.g., incursion, increased tourist visitation) are altering biophysical conditions at many such sites. As a result, conditions at many sites are falling outside the criteria for their original designation. The alternatives are to change the criteria, remove protection from the site, change site boundaries such that the larger or smaller landscape meets the criteria, or manage the existing landscape in some way that reduces the threat. This paper argues for adaptive heritage, an approach that explicitly recognizes changing conditions and societal value. We discuss the need to view heritage areas as parts of a larger landscape, and to take an adaptive approach to the management of that landscape. We offer five themes of adaptive heritage: (1) treat sites as living heritage, (2) employ innovative governance, (3) embrace transparency and accountability, (4) invest in monitoring and evaluation, and (5) manage adaptively. We offer the Australian Wet Tropics as an example where aspects of adaptive heritage currently are practiced, highlighting the tools being used. This paper offers guidance supporting decisions about natural heritage in the face of climate change and non-climatic pressures. Rather than delisting or lowering standards, we argue for adaptive approaches.

**Keywords:** natural heritage; world heritage; protected areas; Outstanding Universal Value (OUV); adaptive heritage; climate change; adaptive management





## 1. Introduction

Protected areas (PA) represent long-term decisions to set aside resources for societal benefit. These benefits include biodiversity and ecosystem services such as aesthetics, carbon sequestration and water quality. Globally, there are more than 200,000 nationally designated protected areas (PAs) representing less than 15% of the earth's surface [1]. Fewer than 1.5% of those PAs are internationally designated, a term that includes natural and mixed World Heritage sites, RAMSAR wetlands and Biosphere Reserves. In fact, UNESCO's Operational Guidelines [2] state that " . . . , the permanent protection of this heritage is of the highest importance to the international community as a whole". These <3000 sites are "the best of the best", locations where attributes are so exceptional that they are seen as worthy of sustained protection for all humankind. Each of these areas has specific characteristics (called Outstanding Universal Value (OUV) in World Heritage terms [2]) that allow the site to qualify as an internationally designated area. Identification and description of those attributes or values initiate long-term site management by orienting the public to the values the site represents [3].

We are using the term heritage to include a wide range of protected places. Each World Heritage site is managed to protect its OUV. Heritage managers strive to balance current and emerging societal attitudes with policies and guidance that reflect the practices



and conditions of the past [3–5]. The operational philosophy within which heritage is managed is to ensure a steady state of the values that originally conveyed heritage [5], despite changing local biophysical and societal conditions. This attempt to capture and retain present or historic value is "postcard heritage" which Kimball [4] characterizes by four axioms or guiding assumptions:

- Heritage itself contains essential qualities that are at risk;
- Designation means that these qualities should be permanently retained;
- These qualities are unrelated to evolution of landscapes or human culture;
- Heritage possesses contingent merit that must be authorized by experts and explained to stakeholders.

The process of listing or designating a landscape as heritage ascribes societal value to the area being listed [5]. These locations are perceived to, but actually may not, have inherent value independent of the viewer [4]. The value, such as the OUV of a World Heritage site, exists to the degree that experts recognize it and assign it to the landscape in question [6]. Many sites might have a particular attribute, but the designated site has a characteristic that has been recognized by experts who led the listing. As custodians of heritage, society's role is to separate the unusual (or unique) landscapes from the common, and to prevent change, to preserve for future generations whatever attributes allow a landscape to qualify as heritage. The argument, which often is explicit in such a designation, is that the past is good, a threat is present that could affect this past state, the future is either bad or unknowable, and the present needs to be protected against that future. However, that judgement is based on the values that are held at the time of inscription. Those very values adopted as reference are mutable, as society changes [7]. There is indeed no constant reference that defines heritage value; value is based on the subjective judgment of experts at the time of inscription.

In the next sections [2–5], we go through typical viewpoints on natural heritage management that are, in our opinion, somewhat or totally outdated. We introduce the concept of adaptive heritage to address the issue of constant change in the condition of OUV. We then present fresh ideas on how to adaptively manage natural heritage in Section 6. In Sections 7 and 8, we highlight innovative examples of adaptive heritage, and we conclude our findings in Section 9.

## 2. Heritage Is Commonly Managed as a Static Resource

Heritage area managers accept, as their mandate, protection of the attributes for which the landscape was designated. Most often, the view is that we must preserve the conditions of the past that led to the current condition, and avoid the losses to that condition that might occur in the future [7]. In Australia, the Burra Charter [8] (Article 3) is often cited as the central guidance for managers and suggests we must do 'as much as necessary but as little as possible' to conserve site attributes. However, that guidance poses a serious challenge to the philosophy and practice of heritage management. All sites are subject to landscape evolution and to a range of pressures, such as climate change, encroachment, and invasive species. Managers need a better understanding of how malleable heritage values are, what forces change those values, and the significance of those changes [9]. As site conditions change, the attributes that supported the listing (e.g., the OUV if it is a World Heritage site) may be lost. That conundrum requires us to ask "If a site loses the attributes of OUV that determined its inscription, is it still World Heritage or heritage quality?" [6,10]. In a current (July 2021) example of that dilemma, the World Heritage Commission considered placing the Great Barrier Reef on the list of Sites in Danger [11]. The Australian government objected, claiming they have spent billions to protect the reef [12]. Clearly, the state party's argument was based on "how hard we tried" rather than the objective qualities of the site and how these may have changed since the original listing in 1980.

Conserving historic values, longing to retain the past and being apprehensive about the future of a site is characteristically a western view [7]. Many African heritage sites (e.g., sacred groves of Nigeria, sacred pools of Katchikally in the Gambia) are viewed as works

in progress, as active sites [13] in which heritage is continually being created (and lost [14]). The difference between static and dynamic heritage is not uniformly seen in Asian or African practices, nor is it uniformly absent from western ones (e.g., [15]). However, the distinction is useful in framing adaptive management as conditions and values change. Nearly all heritage sites are facing some pressure from climate change. Managers must look to the future and frame both reactive and proactive strategies. Recent work [16–18] has shown that most responses are reactive, attempting to replace lost values rather than being proactive about potential future values [9,19]. Proactive management looks to the future, assessing risks to the values of the site and adapting management accordingly.

Landscapes are not static. They have reached their current biophysical condition through landscape evolution, part of which has been influenced by humans [20]. There is also an evolution of societal views and values [21], and a causal interaction between human values and landscapes (i.e., the attributes to which we ascribe value guide our behavior, which in turn influences changes in the landscape [5,22]). Heritage managers recognize that some degree of change in biophysical condition is to be expected. Conceptually, we recognize that variance, and recommend building management strategies to retain heritage sites within the defined "limits of acceptable change" [9]. Here, the concept of a safe operating space [23] is useful in defining the upper and lower boundaries of change that are within the risk appetite, and ensuring that site conditions remain within these boundaries. As climate change (and other influences) alters the conditions of heritage landscapes, we will exceed those limits in many cases. The OUV or other heritage value will move out of the "safe operating space" within which the values are resilient (i.e., bounce back from perturbation). The site will, in effect, transition from one state into another, a condition from which the first state is isolated [5]. The heritage attributes for the site will be lost and in need of redefinition. We will be forced to ask "Is this still valuable as heritage?" and we will be challenged to find credible ways to express our narratives about the history of our heritage sites [24].

## 3. Natural Heritage Is Properly Managed at the Landscape Scale

It has become widespread practice to manage heritage sites as discrete patches in the landscape. That scale makes sense from a practical standpoint. A patch (of any spatial scale) is owned and managed by a bureaucratic entity (e.g., a nation state), and that entity has at least some control over activities within the patch. However, that patch always is influenced by conditions in the surrounding landscape and vice versa. We suggest that effective heritage management must occur at the landscape scale (defined as the zone of influence on the heritage site) and must be adaptive, recognizing changes in both biophysical condition and societal value, an approach we call *adaptive heritage*. In the following sections, we discuss why landscape is the appropriate spatial scale, and discuss what we see as the components of adaptive heritage. We address the conundrum: When a site loses the attributes that make it heritage, do we de-list or lower our standards?

The landscape is the spatial zone of influence (i.e., the greater physical space in which a heritage site resides) [3,9]. The landscape includes the surrounding biophysical as well as the socioeconomic context [3]. Heritage sites themselves vary widely in size, ranging from, for example, Vale de Mai ($0.2$ km$^2$) to the Great Barrier Reef ($3500$ km$^2$) [25]. Human activities (e.g., agriculture, industrialization, urbanization) and natural activities (e.g., erosion, seismicity) in the surrounding landscape often strongly influence the character of the heritage site and may control the degree to which the site meets the criteria that characterize it as heritage. A landscape-based approach to heritage management is increasingly recognized as being essential for effective heritage management [3,19,26,27]. The strength of the landscape approach extends beyond its encompassing scale and includes a focus on flows and linkages among patches [26,27].

One of the characteristics of landscapes that strongly influence heritage management, is that landscapes are dynamic [28]. The present condition of a landscape is a product of natural and anthropogenic influences, and these continue to change through time. Heritage sites are

affected by off-site anthropogenic influences such as biodiversity use, encroachment, invasive species, water management, carbon emissions and climate change, as well as practices that affect the flows of energy and materials. Beyond the biophysical conditions, the values we assign to heritage are themselves dynamic [5]. As Kimball [4] (p. 59) states "Everything arises, persists, and passes away because its temporary existence depends on whatever lineages of phenomena brought it into being, whatever phenomena hold it in place and memory for a time, and whatever phenomena will inevitably cause its undoing".

## 4. Landscapes Are Dynamic Entities

The dynamic nature of landscapes, and the heritage sites within them, requires a management and conservation approach that is sensitive to change [9]. In some ways, this is a wicked problem. We are using ever-changing management strategies to reach ever-changing goals [19]. Heritage sites are biophysical patches, but their values are social constructs. That is, heritage is a complex socio-ecological system (SES). The most effective approach to understanding and managing an SES is through resilience, adaptability and transformability [29]. Transformability, or the ability to cross thresholds into new states, is the latter of those three properties and is central to our ideas of adaptive heritage. As biophysical and socioeconomic conditions change, the SES evolves to follow a new trajectory [30] to a new center of attraction, potentially with attributes that do not meet the original criteria for site designation (i.e., it transitions to a new state).

There are several reasons why the dynamic approach to a heritage landscape SES is useful. Such an approach is science-based but necessarily incorporates human valuation and iterative decision-making [28]. The approach is (or can be) forward looking, using visioning and scenarios to consider future values and future biophysical conditions [5]. An iterative and adaptive approach advances learning, focusing on experimental, untested alternatives [19] from which new strategies and conditions arise [26]. It also acknowledges that we do not, cannot, know everything. Therefore, we act to the best of our knowledge. We live in learning landscapes where we plan, act, monitor, improve, and repeat. The result, when properly implemented, focuses on adaptive management (i.e., learning from trials), on simultaneous attention to multiple objectives and, critically, on stakeholder involvement [28]. As these trials are implemented at the landscape, not the patch scale, stakeholders come from the broader area, increasing the opportunity to define community-wide goals and corrective strategies.

Management of a heritage site (as an SES within a landscape) is an attempt to understand and balance stakeholder needs and values with the landscape trajectory [26,27]. Those stakeholder needs and values must include the criteria used to judge the heritage site when it first was designated [5,7]. The landscape trajectory implies continuity, which differs from integrity. Continuity recognizes the "continuous process of evolving tangible and intangible heritage expressions in response to changing circumstances—in this sense, change is embraced as a part of the continuity" [7] (p. 21). Change in this context includes both the biophysical attributes of the heritage site, and the societal value, the evolving human expression of the ways the site represents something valuable to future generations [15,29,31].

The subset of heritage that we are considering here consists of natural areas that are internationally designated. As such, each heritage site has met the criteria of some designating body, and site management goals are targeted toward, or at least influenced by, those criteria. However, there is a weakness in the ways those goals are framed and implemented for most such heritage sites: they are spatially and temporally constrained. They inadequately consider the sphere of influence of the surrounding biophysical and cultural landscape, or the temporal changes that influence the heritage site. The landscape approach necessarily considers the views and values of a range of stakeholders and forces a consideration of the tradeoffs and co-benefits in that larger spatial area [9]. This broader view of the spatial area and stakeholders advances inclusivity [32] but adds management complexity. That complexity slows decision-making and reduces the probability of con-

sensus, but advances informed and inclusive decisions. Climate change influences all our heritage landscapes, acting as a threat multiplier [14]. As it does so, the need for informed and inclusive decisions increases. Sustaining heritage landscapes is not a linear process, especially as climate change increases climatic variance. That variance creates windows of risk and opportunity that may cause a phase shift from one stability regime to another [30].

## 5. As a Dynamic Resource, Natural Heritage Is always Changing

Heritage conservation, by necessity works, with those windows of risk and opportunity. Our landscapes, as well as the societal values that establish our goals and benchmarks, are continually in a state of flux. Conservation is management of change or risk, compatible with predefined objectives in an ever-evolving world [33]. In this sense, compatible means changes that retain the heritage values. The framework of continual change and, therefore, limits of acceptable change within a safe operating space, poses significant challenges to our (implicit or explicit) goal of maintaining a constant value, and continuing to meet the criteria for site OUV [10]. Heritage management has traditionally involved setting aside areas (such as RAMSAR wetlands or Biosphere Reserves or World Heritage sites) and attempting to prevent change by managing threats. However, that management approach clearly does not consider either climate change [10], or anthropogenic changes in the surrounding landscape. Managers will be able to affect some but not all those external influences. As those external influences change the site, managers will be faced with conditions that do not meet the OUV (or other listing criteria). That requires a decision: Do we lower the bar for eligibility or decide the site no longer represents heritage [10]? Do we accept a shifting baseline for heritage? These questions require a societal answer, and lie within the philosophical domain of acceptance that all does not remain constant in this world.

On the surface, it seems that "management of change" [9,21,34] is the operational philosophy that should guide management. In that sense, heritage becomes a process bounded by the biophysical landscape on one hand and societal values on the other [4,35]. That process has a momentum, a continual change through time [27], that constrains and empowers management. Heritage conservation becomes an attempt to balance tradeoffs, accommodating some magnitude of change, some deviance from original condition, yet retaining some semblance of the values society holds for the site [5,10]. An example of such balancing comes from iSimangaliso Wetland Park (KwaZulu-Natal, South Africa). Interactions among the national government, local government and residents have resulted in a trajectory of values, a changing and often contentious expression through time of the values to be retained in the landscape [36].

*Adaptive heritage* is the suite of views and practices that allows us to manage that momentum. Viewing a heritage site as an SES recognizes the interaction between the biophysical and the social aspects of the broader landscape and where the site sits in that interactive space. An adaptive approach to SESs recognizes that the system learns, in the sense that condition and behavior today are a function of conditions and behaviors in the past [30,36]. That recognition frames the heritage site today as having a place on a trajectory, rather than a fixed condition. Examples of that trajectory abound. Pastoral heritage, as seen in many parts of North Africa, is maintained only by accepting that both climates and social conditions are continually in a state of flux [14]. The current landscapes of the Australian Wet Tropics are a function of centuries of human manipulation through fire [10,37]. Kimball [4] terms this "regenerative conservation", movement to a new state rather than the static view of conserving fixed conditions. However, regenerative conservation risks encountering shifting baselines, and the need to change heritage criteria to encompass new site conditions.

## 6. Ideas for Approaching Adaptive Heritage

In this final section, we offer five themes that we suggest would advance adaptive heritage, and a site-specific example where some of these ideas currently are practiced to some degree. Rather than advocating for radical change, we suggest that a reflective

approach, gradually increasing the degree to which these ideas are implemented, will increase resilience and advance adaptive management. We begin by considering the fundamental question considered by many and raised most recently by Weber et al. [10]: As site conditions change and a site fails to meet its heritage criteria (or loses its OUV), do we lower the standards or delist the site? Our argument is that standards are not meaningful if they are adjusted to meet the situation. That is, we do not lower the standards. Rather, governing bodies (e.g., the World Heritage Center) should continue using "Sites in Danger" and other descriptors, perhaps including new judgments such as "Sites in Climatic Transition" [5]. Further, to be useful, a standard must be judged against objective measures. Returning to the Great Barrier Reef example, if on-site conditions do not meet the designation criteria, the site is objectively In Danger. However, that is not, and should not, be the whole story. A site In Danger, with a national steward investing very heavily in protection and management, has a different trajectory than a site without a national champion. It is the responsibility of the oversight bodies to recognize the difference between those two situations. Implementing the principles of adaptive heritage in a Site in Danger will create a positive momentum toward reduced threat.

### 6.1. Five Themes of Adaptive Heritage

#### 6.1.1. Adaptive Management

Adaptive management involves the cycle of framing (and communicating) goals; implementing practices that, to the best of our knowledge, will meet those goals; monitoring the response of the system to those interventions; evaluating (and communicating) results; revising the goals and practices; and starting again. External factors such as climate change and social practices (e.g., tourism, incursion) cause changes in our practices and their success [38]. Climate change specifically will strongly affect many heritage sites, causing changes in species composition and driving the need for adaptive conservation strategies [39]. As those changes occur, sites will fail to meet some of the criteria for heritage listing. Seekamp and Jo [5] have suggested a new, perhaps adaptive, term for such conditions: World Heritage Sites in Climatic Transformation, a way to frame the cultural values and landscape conditions derived from a WH site (or other protected area).

#### 6.1.2. Transparency and Accountability

Heritage conservation is always managed at the intersection of biophysical properties and societal value. Stakeholders, from the local to the global, supposedly represent that valuation, but experts and managers translate that valuation into action. As both landscape conditions and valuation evolve, heritage managers have an increasing responsibility to communicate clearly and frequently with stakeholders. That clear communication is essential if we expect stakeholders to act within their spheres of influence to support this reflective approach of evolving landscapes and evolving societal value As such, interventions need to have transparent assumptions, precise monitoring [26] and transparent reporting. Heritage sites (and their managers) need to sustain a sense of accountability among stakeholders, supporting experimentation and adaptive management, and recognizing that precise outcomes are difficult to predict. Heritage managers are the focal point of changes because their experience and responsibility provide direct and actionable information about the landscape they manage [32]. In the case of the recent scientific advice to place the Great Barrier Reef as "In Danger", the process was affected by significant political lobbying by the Australian Government in the weeks leading up to the World Heritage Council decision. Political pressure is but one facet of the complex SES under which World Heritage sites are designated. Scientific evidence is at the fore of assessments as to ongoing status and should be defended against lobbying from state parties.

#### 6.1.3. Monitoring and Evaluation

An adaptive approach requires us to assess performance and adjust our actions. Similarly, accountability consists of framing and publicizing goals, actions, and evaluations.

Transparency with stakeholders builds credibility and a sense of shared understanding. All those aspects of evaluation require information, which necessitates monitoring. Collectively, monitoring and evaluation allow managers to set priorities, experiment, and determine the effectiveness of various actions [32]. We argue that heritage is best managed at the landscape scale, a wicked problem that requires nuanced evaluation metrics within and surrounding the site. Monitoring data should be framed to inform management action. For example, species attributes (e.g., richness, diversity) are the most common metrics for evaluating natural protected areas [40]. Recent work [41] provides guidance on a large variety of biophysical variables that can be collected remotely and can provide fine temporal and spatial scale information. However, the purpose of monitoring and evaluation is to inform management, inform a broad stakeholder group, and provide transparency. That requires much more than objective measures of biodiversity, including aspects of composition, complexity, and abundance, particularly of species upon which OUV is based. This can be costly, and surrogates that can be remotely sensed will be useful [42]. Natural heritage sites are SESs, so monitoring must include the greater landscape. Furthermore, stakeholder characteristics (e.g., economics, cultural background) affect their interest in various performance indicators [32], necessitating a diverse approach to metric selection.

### 6.1.4. Innovative Governance

Historically, many PAs were managed by a team of technically skilled professionals who were given a charge (e.g., manage an appropriate balance between tourism and conservation). Understanding that the heritage site is indeed part of a larger landscape with its attendant range of stakeholders requires the framing of goals and expectations in a broader context. All heritage areas have management staff, who are responsible for the resources at the site. However, advisory councils and other shared governance approaches empower co-management, shared goals, and a sense of community, both among those responsible for site management and within the broader community in the landscape [32]. Given that many sites are, or were, home to First Nations people, their role in governance is paramount, and needs to reflect indigenous rather than strictly Western approaches to governance. That broader definition of stakeholders and goals is well done at some locations (e.g., Australian Wet Tropics, Glacier National Park) but, to date, under-addressed in many locations. Shared governance and increased transparency will be new and difficult for some expert managers. There will necessarily be tradeoffs and the sustaining of the objective biophysical attributes that supported the original designation may be compromised.

### 6.1.5. Living Heritage

Most heritage sites are defined based on a series of values present at the time of designation (e.g., OUV in a Word Heritage context) and are managed in a static way to maintain those values. In that sense, the value is defined by past events. The implicit statement is as follows: historic conditions have produced a condition we find attractive, and future changes away from that attractive state are to be avoided. Authenticity, a core component of site valuation (e.g., [2]), suggests that site conditions must remain close to some defined or imagined prior condition. In contrast, living heritage is a process-based approach in which change through time is an inherent part of a site's value [7]. Living heritage suggests that authenticity is defined by the evolving intersection of communities (i.e., stakeholders) and site biophysical attributes [7]. Living heritage sites are seen as continually being in transition, a process that might be termed "transformative continuity" [5]. The attribute that makes a living heritage site "authentic" is the degree to which there is congruence between community values and heritage conditions. Rather than seeking to avoid change, managers of living heritage sites seek to advance that congruence. In doing so, managers must separate influences that can be managed (e.g., pests, fire) and ones that cannot (e.g., climate change).

## 7. Examples of Innovation

Many authors have suggested and/or demonstrated approaches and tools that could advance adaptive heritage (e.g., [39,41]). We close this paper with comments on two promising tools, and an example of a natural heritage site that demonstrates innovation. These are intended to provoke thought and discussion rather than offer fait accompli.

### 7.1. Heritage Tourism

Part of the expectation for heritage sites is representing values important to the general public, often through tourism. However, tourism has to be managed if it is not to be a threat to the OUV, or if the OUV is to be dynamic in the face of increasing tourism to the site. Heritage conservation is an expensive management practice. It involves setting aside lands for conservation, and investing time, energy, and resources in managing those lands to sustain the accepted values. Many sites are in the thrall of tourism, because it is a significant value for heritage sites. Tourism revenue sustains many heritage management practices. Global tourism was valued at $2.9 Trillion (USD) in 2019 [43], and heritage tourism accounts for approximately half of that value [42]. An economic force of that magnitude drives many decisions, including the risk that tourism becomes so financially attractive it destroys the site. There are opportunities to influence those decisions. At the scale of the individual traveler, tools such as the Climate Footprints of Heritage Tourism are available online (in that case, as a publicly available ArcGIS StoryMap) [42]. In contrast, at the scale of the individual heritage site, there is a positive opportunity for management in support of tourism values, but counter-pressure to ensure that the site is managed for an inclusive range of stakeholders. Ecotourism is seen as compatible with many natural heritage sites but is often expensive and exclusive. That raises a conflict requiring attention: how do we sustain low-density, low-impact tourism and meet societal goals of inclusivity [44]?

### 7.2. Regenerative Conservation

Kimball [3] has suggested that heritage conservation would be advanced through the practice of "empty heritage". In his view, all heritage sites are devoid of any inherent quality. Rather, the qualities we ascribe to a heritage site are a function of the values we hold, and those values change with time and among stakeholders. He further suggests that careful examination of those values and the stories supporting them can lead to regenerative conservation. That challenges the definition and many of the applications of heritage conservation. Weber et al. [10], among others, have shown that the heritage values of the Australian Wet Tropics are indeed the product of thousands of years of human manipulation. Archer et al. [45] discuss species that have been absent from certain lowland landscapes since the late Pleistocene; however, those lowlands are returning to pre-Pleistocene conditions due to climate change. If a species historically present, but absent for >10,000 years is introduced to the lowlands, will the species have heritage value? Burney [46] and others have spent two decades restoring Makauwahi Cave in Kauai, Hawaii, intending to reconstitute the local ecosystem as it existed before the arrival of humans, as much as is possible. Is such a reconstituted landscape of heritage value? If not, what aspects of heritage definition are not met?

## 8. An Example of Adaptive Heritage in Practice

The Wet Tropics World Heritage Area (WTWHA) rainforest was listed by UNESCO for all four natural criteria in 1988 and was recognised on the Australian National Heritage list for its cultural values in 2012 [10]. The WTWHA contains rainforests that have existed on these mountain ranges for over 130 million years. The site also is home to one of the world's oldest living cultures: Rainforest Aboriginal Peoples (RAP) have been living here for at least 5000 years [47]. Before European settlement, the Wet Tropics rainforests were one of the most diversely populated areas of Australia, and the only place where Australian Aboriginal people permanently inhabited a tropical rainforest environment [48]. Rainforest Aboriginal People developed a specialised and distinctive cultural heritage as well as

traditional food gathering, processing, and land management techniques, shaping the soil, as well as animal and plant species' composition and distribution. Landscape management through fire and cultivation shaped the natural values upon which the WTWHA was gazetted in the 1980s, highlighting the role of socio-ecological systems in engaging the natural heritage and the dynamic nature of the landscapes. The challenge remains that the World Heritage listing favoured management policies that restricted RAP access to its forest environments, favouring botanical novelty and evolutionary trajectories over human history [36]. Recent, more enlightened approaches to the Wet Tropics Management Authority have recognised the important role that RAP have played, and do play, in these dynamic landscapes [10]. The Board of the Wet Tropics Management Authority also now has two positions specifically for a male and female RAP, chosen by the State and Federal Governments. Co-management of the WTWHA by the Australian government and the Aboriginal Peoples has several attributes of adaptive heritage. Management is adaptive, meaning that it is co-designed with RAP and local communities, and the site is clearly seen as living heritage. Actively engaged citizen and management groups communicate frequently, using monitoring and evaluation to advance transparency and accountability [10].

## 9. Conclusions

We suggest that heritage conservation be advanced by an adaptive approach. Our term adaptive heritage recognizes that management must pursue two moving targets: biophysical change (on-site and in the landscape) and societal value. We have positioned this argument in the literature that guides heritage valuation, arguing that protected areas are best managed in a landscape context and that both the biophysical and sociocultural aspects of that landscape are subject to continual change. Heritage sites are not cocooned from those changes. We have offered five themes of an adaptive approach that we suggest would improve the relationship between heritage conservation as a practice and the society (i.e., the societal valuation) that supports such conservation. We also offer guidance, supporting decisions about natural heritage in the face of climate change and non-climatic pressures. As examples, we cite the Great Barrier Reef, where adaptive heritage would avoid delisting or lowering standards, and the Wet Tropics, where we see adaptive approaches currently being practiced.

**Author Contributions:** J.P., I.J.G. contributed equally to all aspects of this manuscript. Both authors have read and agreed to the published version of the manuscript.

**Funding:** This research received no external funding.

**Institutional Review Board Statement:** Not applicable; this study did not involve humans or animals.

**Informed Consent Statement:** Not applicable.

**Data Availability Statement:** Not applicable.

**Acknowledgments:** We would like to thank 2 reviewers for their constructive comments on the manuscript. Their feedback has been valuable in tightening up the arguments of our case.

**Conflicts of Interest:** The authors declare no conflict of interest.

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
