# Peer review of "Adaptive Heritage: Is This Creative Thinking or Abandoning Our Values?"

_climate, doi:10.3390/cli9080128_

Round 1
Reviewer 1 Report
In sections 8 and 9 examples of adaptive heritage is presented. However, it is not clear which themes are actually implemented in these examples. Reviewer would request the authors to explain which themes were implemented and if possible discuss the challenges and success of the themes for each example.
Author Response
Reviewer 1
- We have added the requersted information

Reviewer 2 Report
Comments on Adaptive heritage: Is this creative thinking or abandoning our values?
- In the last paragraph of the Introduction, the authors should clearly mention the weakness point of former works (identification of the gaps) and describe the novelties of the current investigation to justify us the paper deserves to be published in this journal.
- Cite this recent useful paper on the importance of heritage studies to improve the literature and to show the importance of your work:
Hydrometeorology: Review of Past, Present and Future Observation Methods
- Discuss more the ideas for approaching adaptive heritage.
- At the end of the manuscript, explain the implications and future works considering the outputs of the current study.
Author Response
Reviewer 2
- In the last paragraph of the Introduction, the authors should clearly mention the weakness point of former works (identification of the gaps) and describe the novelties of the current investigation to justify us the paper deserves to be published in this journal.
- As suggested, we have added such text
- Cite this recent useful paper on the importance of heritage studies to improve the literature and to show the importance of your work: Hydrometeorology-Review of Past, Present and Future Observation Methods
- The suggested paper is not relevant to this work
- Discuss more the ideas for approaching adaptive heritage
- The discussion of these ideas has been expanded
- At the end of the manuscript, explain the implications and future works considering the outputs of the current study
- This has been included

Reviewer 3 Report
General comments to the authors
The subject of the paper is of interest to many readers, as changes presented especially via climate change are rapid and our perspectives on nature protection are still based on more static world. We desperately need more concrete examples of adaptive nature management, and also examples on solutions to the many problems this adaptiveness will present. This paper can partly shed some light to the issue, but I encourage the authors to present their findings in even more concrete way (especially their example on Australian Wet Tropics). Otherwise, the paper falls into the common category of repeating “this is how we should orient to the issue” and not giving any concrete advice on how to actually proceed. Bearing this in mind, my comments about the text are otherwise mostly about 1) omitting repetition and 2) re-orienting the structure and order of presenting (“storyline”), not so much on the content per se. Even though re-structuring and concretizing the text might take some effort, I evaluate this as a “minor revision”.
Specific comments to the authors
About the title and keywords: As ”adaptive heritage” is a totally new term (at least for me) it is important to add to the keywords “adaptive management” which is more common.
About the titles in parts 2-5: I would love to have a clearer understanding if these statements that you make in the titles are something that you criticize or stand for (now there is both, compare titles 2 and 3). You could state this at the very end of the introduction, giving a reading instruction in the same time. E.g.: “In the next chapters (2-5), we will go through typical viewpoints on natural heritage management that are, in our opinion, somewhat or totally outdated. We then present fresh ideas how to adaptively manage natural heritage in chapters 6 and 7. In chapters 8 and 9 we highlight innovative examples of adaptive heritage, and we conclude our findings in chapter 10.” There is something like this in lines 143-147, but I would like to see it earlier in the text, given that part 2 is still going to be there (and not fused into introduction).
Line 31: I am used to think that biodiversity underlies ecosystem services, and thus I would rephrase the sentence: “These benefits include biodiversity and ecosystem services such as aesthetics…”
Line 38: Number 2) here refers to a reference (2), right?
Line 47: Degradation is change. NEGATIVE change! PAs usually are designated in a way that allows for evolutionary changes (in fact, this is one of the basic reasons to protect a site, to allow the populations within to breed, spread and finally evolve). So not all change is considered inherently bad when designating a PA. The sentence as it stands now is too simplifying, please rephrase.
Lines 57-71: If a reader does not have profound knowledge on World Heritage sites or OUVs, they are left to wonder about the word “value” here. Please elaborate and/or give some common examples here (or, if they are given soon in the text below, point to this). Are we talking mostly about ecological values? Aesthetics? Or both, or something else? This is especially important as you go on about how these values are “in the eye of the expert” and how they tend to get “fixed” in a process of setting up the heritage site.
Line 77: I would suggest: “In Australia, the Burra…”. I do understand the universality of the principle, but your reference is so very Australian that it seems a little funny to think that PA managers around the world would know it. Other option is to find more common references here.
Line 82-83: “Managers need a better understanding…” This is a very good sentence, right on the point!
Lines 92-101: Nothing wrong in this paragraph, except that I have a faint feeling that I have read this content before. Please check if this is repetitive e.g. with lines 42-49 or partly with 57-71. I think this paragraph is well written, so maybe consider keeping this one and omitting the text before where the same was said more bluntly.
Line 106: I would put the reference 17 within the same parentheses with “and lost”. Same elsewhere in the text (but maybe the paper has a different instruction?).
Lines 115-125: See my comment on line 47 above. I feel that this (nice) paragraph could be presented earlier in the text, more like an introduction or presenting the status quo. This gives credit to the PA managers that they are not dummies that just want to store nature in a museum (to put very bluntly, this was the feeling I got from the intro). What is new, is the RATE at which things are changing now and in the near future. That is the big challenge we need to face and we need new tools to cope with. See the next comment immediately below.
Lines 125-133: Very well put, here is the main challenge in a nutshell. I am not sure if this bit can be presented a bit further from the lines 115-125, but if it could, I would love to see the lines 115-125 maybe at the very beginning of the part 2, setting the scene as it is, and then moving to compare Western and African/Asian ways to see the change, and maybe finally coming back to this (most important) bit, pointing the need to change our viewpoint on PA management all over the world. So the contents of the text is mostly fine, but I would have another look at the storyline in parts 1 (intro) and 2, and remove repetitions to make it even smoother. You could also think about if part 2 is actually needed or should it be fused to be part of introduction?
Lines 121-123: There seems to be something (maybe a verb) missing from this sentence.
Line 142: How does adaptive heritage differ from the term adaptive management? Does it have something to do about his spatial scale issue, or is there another reason to use a new term?
Line 210: These constraints are not very striking to me, more so they seem practical (like that there is a limit to the spatial and temporal scales of operation). Maybe elaborate or clarify what is the striking weakness here, or choose somewhat milder expression.
Lines 225-242: Again, I feel repetition. Please check and omit parts that repeat this “static versus dynamic” issue. It is not a bad outcome if the text shortens in the process.
Lines 243-249: This is more interesting: trying to concretize how does this new management methodology look like. I long for real-life examples already… The same feeling goes with the wider landscape management “method” – maybe reconsider showing examples earlier? Like in lines 256-262 you so nicely do.
Line 250: See my comment on line 142. Is adaptive heritage something that falls into a wider category of adaptive management? Is it more specific or does it encompass totally new elements compared to adaptive management?
Lines 264-284: In my opinion, this section does not need its own title, it is more like a small intro of the next section (and thus should go under its title “Five themes…”). The way you start it already shows it: “In this final section…” (is it final, after all there is now 10 sections?). I like these little reading instructions, so please keep it. I also like the concrete suggestions made here (starting from line 273), but are they in a right place? Thinking that this bit would belong under the title in section 7, I make some suggestions below.
Lines 289-291: These sentences, at this point, can clearly be omitted (we certainly know this by now!). Please start straight with “Adaptive management involves…”.
Line 302: Here you have the same reference as in line 276, and if you read the texts, they tell the same story. What I miss, though, is your own viewpoint on the matter, and that you present in the earlier text bit (274-284). It would go nicely here, at least partly (see the next comment).
Lines 305-316: Thinking back to the Great Barrier Reef example, could you use it here to concretize the issue of transparency and accountability? One aspect of accountability is honesty, and I feel you are underlining that when you state in lines 273-274 that standards should stay in place (I fully agree!), and that In Danger -category is appropriate even though there was a political pressure against it. My point is: to stay open and listen to the stakeholders (politicians being one stakeholder group too) cannot mean that we throw out all the original standards on why the area was worthwhile protecting in the first place. Surely monetary efforts matter something, but it is the ecological state (and values, I presume!!) that we want to conserve, and thus those are the ones that should be measured and looked at first and foremost.
Lines 329-330: Much more – like what? Please provide some examples of other indicators than those related to biodiversity (e.g. maybe you mean ecosystem service indicators?). A point to mention here would be that we can collect quite a lot of information using GIS and satellite-based methods (check e.g. Skidmore et al. 2021: Priority list of biodiversity metrics to observe from space).
Lines 335-366: As a reader, although I understand the points made in these two subsections, there are many questions that immediately rise and which you do consider or present here. Who makes the important decisions on accepting or denying a change in the management goals of the site? How can we make sure that power issues are properly dealt with in advisory councils or stakeholder boards (a lot of literature on this e.g. in the context of co-production of knowledge)? How can we assure that the original (ecological) viewpoint is not totally lost in the process of accepting change (say, so that a wooded PA wouldn’t suddenly become clearcut because a strong stakeholder group values timber revenues more than the standing forest)? I am not saying you need to have answers to all these questions (and more), but it would be honest to admit that these questions exist and need to be considered in adaptive heritage.
Lines 367-409: In the small intro before the two examples you state that these are “promising tools to advance adaptive heritage”. However, in both cases, you end up with open questions (much like I advised you to do in the comment above) leaving me wonder the usefulness of these examples. I get that there are pros and cons (and open questions), it is just that the text to me does not deliver what it promises. Luckily, the text can be modified to do so. There are several options, and first I would check if these examples from literature could be presented earlier in the text (maybe just a little bit shortened) to back up your statements or to elaborate them. See e.g. my comment on lines 243-249. This option would then leave more space to section 9, where the real-world example is presented (as section 8 would disappear). Second option is to modify the small intro here, maybe end it up with a sentence like: “However, although promising, both oh these approaches have open questions.” Although the second option is easier, I encourage you to ponder on the first one too.
Lines 415-418: These two sentences seem to require a reference.
Line 432: Check the spelling in the word “Aboriginal”.
Line 433: Please elaborate how the management is adaptive (how does it show in practice) and if you have an example of living heritage, this would be a perfect place to concretize it some more. This section could be more than one (long) paragraph in length – this really should be the jewel of the paper!
Author Response
Reviewer 3
- The subject of the paper is of interest to many readers, as changes presented especially via climate change are rapid and our perspectives on nature protection are still based on more static world. We desperately need more concrete examples of adaptive nature management, and also examples on solutions to the many problems this adaptiveness will present. This paper can partly shed some light to the issue, but I encourage the authors to present their findings in even more concrete way (especially their example on Australian Wet Tropics). Otherwise, the paper falls into the common category of repeating “this is how we should orient to the issue” and not giving any concrete advice on how to actually proceed. Bearing this in mind, my comments about the text are otherwise mostly about 1) omitting repetition and 2) re-orienting the structure and order of presenting (“storyline”), not so much on the content per se. Even though re-structuring and concretizing the text might take some effort, I evaluate this as a “minor revision”.
- We appreciate the comments, thank you
- Specific comments to the authors
- About the title and keywords:
- As ”adaptive heritage” is a totally new term (at least for me) it is important to add to the keywords “adaptive management” which is more common.
- We have added this to the keywords
- About the titles in parts 2-5:
- I would love to have a clearer understanding if these statements that you make in the titles are something that you criticize or stand for (now there is both, compare titles 2 and 3). You could state this at the very end of the introduction, giving a reading instruction in the same time. E.g.: “In the next chapters (2-5), we will go through typical viewpoints on natural heritage management that are, in our opinion, somewhat or totally outdated. We then present fresh ideas how to adaptively manage natural heritage in chapters 6 and 7. In chapters 8 and 9 we highlight innovative examples of adaptive heritage, and we conclude our findings in chapter 10.” There is something like this in lines 143-147, but I would like to see it earlier in the text, given that part 2 is still going to be there (and not fused into introduction).
- Thanks; text added at the end of section 1
- Line 31:
- I am used to think that biodiversity underlies ecosystem services, and thus I would rephrase the sentence: “These benefits include biodiversity and ecosystem services such as aesthetics…”
- We have made that change
- Line 38:
- Number 2) here refers to a reference (2), right?
- We have made that change
- Line 47:
- Degradation is change. NEGATIVE change! PAs usually are designated in a way that allows for evolutionary changes (in fact, this is one of the basic reasons to protect a site, to allow the populations within to breed, spread and finally evolve). So not all change is considered inherently bad when designating a PA. The sentence as it stands now is too simplifying, please rephrase.
- This sentence has been rephrased as suggested
- Lines 57-71:
- If a reader does not have profound knowledge on World Heritage sites or OUVs, they are left to wonder about the word “value” here. Please elaborate and/or give some common examples here (or, if they are given soon in the text below, point to this). Are we talking mostly about ecological values? Aesthetics? Or both, or something else? This is especially important as you go on about how these values are “in the eye of the expert” and how they tend to get “fixed” in a process of setting up the heritage site.
- This has been revised to be more clear
- Line 77:
- I would suggest: “In Australia, the Burra…”. I do understand the universality of the principle, but your reference is so very Australian that it seems a little funny to think that PA managers around the world would know it. Other option is to find more common references here.
- Jim’s response
- Agree; we have made that change
- Line 82-83:
- “Managers need a better understanding…” This is a very good sentence, right on the point!
- Thank you for the comment
- Lines 92-101:
- Nothing wrong in this paragraph, except that I have a faint feeling that I have read this content before. Please check if this is repetitive e.g. with lines 42-49 or partly with 57-71. I think this paragraph is well written, so maybe consider keeping this one and omitting the text before where the same was said more bluntly.
- Moved and edited
- Line 106:
- I would put the reference 17 within the same parentheses with “and lost”. Same elsewhere in the text (but maybe the paper has a different instruction?).
- No, that would not be appropriate. As we state it, “... heritage is continually being created (and lost) (17)” means that ref 17 said both. If we put the ref inside the parens, that would mean we are saying created, but ref 17 is saying lost.
- Lines 115-125:
- See my comment on line 47 above. I feel that this (nice) paragraph could be presented earlier in the text, more like an introduction or presenting the status quo. This gives credit to the PA managers that they are not dummies that just want to store nature in a museum (to put very bluntly, this was the feeling I got from the intro). What is new, is the RATE at which things are changing now and in the near future. That is the big challenge we need to face and we need new tools to cope with. See the next comment immediately below.
- We have moved the paragraph to be more clear
- Lines 125-133:
- Very well put, here is the main challenge in a nutshell. I am not sure if this bit can be presented a bit further from the lines 115-125, but if it could, I would love to see the lines 115-125 maybe at the very beginning of the part 2, setting the scene as it is, and then moving to compare Western and African/Asian ways to see the change, and maybe finally coming back to this (most important) bit, pointing the need to change our viewpoint on PA management all over the world. So the contents of the text is mostly fine, but I would have another look at the storyline in parts 1 (intro) and 2, and remove repetitions to make it even smoother. You could also think about if part 2 is actually needed or should it be fused to be part of introduction?
- We have moved and revised the text to be more clear
- Lines 121-123:
- There seems to be something (maybe a verb) missing from this sentence.
- Well spotted; we have made the correction
- Line 142:
- How does adaptive heritage differ from the term adaptive management? Does it have something to do about his spatial scale issue, or is there another reason to use a new term?
- We have expanded our definition to clarify the distinction: The intent of the term is to incorporate adaptive management with the implied threshold of the criteria for designation. We are intentionally suggesting that we must manage for a moving target.
- Line 210:
- These constraints are not very striking to me, more so they seem practical (like that there is a limit to the spatial and temporal scales of operation). Maybe elaborate or clarify what is the striking weakness here, or choose somewhat milder expression.
- We have modified the text to further explain what we see as weaknesses
- Lines 225-242:
- Again, I feel repetition. Please check and omit parts that repeat this “static versus dynamic” issue. It is not a bad outcome if the text shortens in the process.
- We have modified the text to reduce repetition and increase clarity
- Lines 243-249:
- This is more interesting: trying to concretize how does this new management methodology look like. I long for real-life examples already… The same feeling goes with the wider landscape management “method” – maybe reconsider showing examples earlier? Like in lines 256-262 you so nicely do.
- We have added examples of the ideas to demonstrate the points being made.
- Line 250:
- See my comment on line 142. Is adaptive heritage something that falls into a wider category of adaptive management? Is it more specific or does it encompass totally new elements compared to adaptive management?
- See response to comment on line 142. Removed adaptive management to avoid confusion
- Lines 264-284:
- In my opinion, this section does not need its own title, it is more like a small intro of the next section (and thus should go under its title “Five themes…”). The way you start it already shows it: “In this final section…” (is it final, after all there is now 10 sections?). I like these little reading instructions, so please keep it. I also like the concrete suggestions made here (starting from line 273), but are they in a right place? Thinking that this bit would belong under the title in section 7, I make some suggestions below.
- We agree; this has been resolved
- Lines 289-291:
- These sentences, at this point, can clearly be omitted (we certainly know this by now!). Please start straight with “Adaptive management involves…”.
- We agree; this has been resolved
- Line 302:
- Here you have the same reference as in line 276, and if you read the texts, they tell the same story. What I miss, though, is your own viewpoint on the matter, and that you present in the earlier text bit (274-284). It would go nicely here, at least partly (see the next comment).
- This has been addressed through our response to next reviewer comment
- Lines 305-316:
- Thinking back to the Great Barrier Reef example, could you use it here to concretize the issue of transparency and accountability? One aspect of accountability is honesty, and I feel you are underlining that when you state in lines 273-274 that standards should stay in place (I fully agree!), and that In Danger -category is appropriate even though there was a political pressure against it. My point is: to stay open and listen to the stakeholders (politicians being one stakeholder group too) cannot mean that we throw out all the original standards on why the area was worthwhile protecting in the first place. Surely monetary efforts matter something, but it is the ecological state (and values, I presume!!) that we want to conserve, and thus those are the ones that should be measured and looked at first and foremost.
- Revolved; we added “In the case of the recent scientific advice to place the Great Barrier Reef as “in danger”, the process appears to have been affected through significant political lobbying by the Australian Government in the weeks up to the decision made by the World Heritage Council. Politics is but one facet of the complex socio-ecological system under which World Heritage sites are assessed for their appointment. Scientific evidence is at the fore of assessments as to ongoing status and such be defended against lobbying from states.”
- Lines 329-330:
- Much more – like what? Please provide some examples of other indicators than those related to biodiversity (e.g. maybe you mean ecosystem service indicators?). A point to mention here would be that we can collect quite a lot of information using GIS and satellite-based methods (check e.g. Skidmore et al. 2021: Priority list of biodiversity metrics to observe from space).
- We added “including aspects of composition, complexity, and abundance, particularly of species on which OUV is based. This can be costly and surrogate that can be remotely sensed measures may have to be used.”
- Lines 335-366:
- As a reader, although I understand the points made in these two subsections, there are many questions that immediately rise and which you do consider or present here. Who makes the important decisions on accepting or denying a change in the management goals of the site? How can we make sure that power issues are properly dealt with in advisory councils or stakeholder boards (a lot of literature on this e.g. in the context of co-production of knowledge)? How can we assure that the original (ecological) viewpoint is not totally lost in the process of accepting change (say, so that a wooded PA wouldn’t suddenly become clearcut because a strong stakeholder group values timber revenues more than the standing forest)? I am not saying you need to have answers to all these questions (and more), but it would be honest to admit that these questions exist and need to be considered in adaptive heritage.
- We have expanded the discussion here to add examples and clarify
- Lines 367-409:
- In the small intro before the two examples you state that these are “promising tools to advance adaptive heritage”. However, in both cases, you end up with open questions (much like I advised you to do in the comment above) leaving me wonder the usefulness of these examples. I get that there are pros and cons (and open questions), it is just that the text to me does not deliver what it promises. Luckily, the text can be modified to do so. There are several options, and first I would check if these examples from literature could be presented earlier in the text (maybe just a little bit shortened) to back up your statements or to elaborate them. See e.g. my comment on lines 243-249. This option would then leave more space to section 9, where the real-world example is presented (as section 8 would disappear). Second option is to modify the small intro here, maybe end it up with a sentence like: “However, although promising, both of these approaches have open questions.” Although the second option is easier, I encourage you to ponder on the first one too.
- Resolved
- Lines 415-418:
- These two sentences seem to require a reference.
- Two references added
- Line 432:
- Check the spelling in the word “Aboriginal”.
- Done
- Line 433:
- Please elaborate how the management is adaptive (how does it show in practice) and if you have an example of living heritage, this would be a perfect place to concretize it some more. This section could be more than one (long) paragraph in length – this really should be the jewel of the paper!
- This has been clarified, meaning that it is co-designed with RAP and local communities
- As a reader, although I understand the points made in these two subsections, there are many questions that immediately rise and which you do consider or present here. Who makes the important decisions on accepting or denying a change in the management goals of the site? How can we make sure that power issues are properly dealt with in advisory councils or stakeholder boards (a lot of literature on this e.g. in the context of co-production of knowledge)? How can we assure that the original (ecological) viewpoint is not totally lost in the process of accepting change (say, so that a wooded PA wouldn’t suddenly become clearcut because a strong stakeholder group values timber revenues more than the standing forest)? I am not saying you need to have answers to all these questions (and more), but it would be honest to admit that these questions exist and need to be considered in adaptive heritage.
- Much more – like what? Please provide some examples of other indicators than those related to biodiversity (e.g. maybe you mean ecosystem service indicators?). A point to mention here would be that we can collect quite a lot of information using GIS and satellite-based methods (check e.g. Skidmore et al. 2021: Priority list of biodiversity metrics to observe from space).
- Thinking back to the Great Barrier Reef example, could you use it here to concretize the issue of transparency and accountability? One aspect of accountability is honesty, and I feel you are underlining that when you state in lines 273-274 that standards should stay in place (I fully agree!), and that In Danger -category is appropriate even though there was a political pressure against it. My point is: to stay open and listen to the stakeholders (politicians being one stakeholder group too) cannot mean that we throw out all the original standards on why the area was worthwhile protecting in the first place. Surely monetary efforts matter something, but it is the ecological state (and values, I presume!!) that we want to conserve, and thus those are the ones that should be measured and looked at first and foremost.
- Here you have the same reference as in line 276, and if you read the texts, they tell the same story. What I miss, though, is your own viewpoint on the matter, and that you present in the earlier text bit (274-284). It would go nicely here, at least partly (see the next comment).
- These sentences, at this point, can clearly be omitted (we certainly know this by now!). Please start straight with “Adaptive management involves…”.
- In my opinion, this section does not need its own title, it is more like a small intro of the next section (and thus should go under its title “Five themes…”). The way you start it already shows it: “In this final section…” (is it final, after all there is now 10 sections?). I like these little reading instructions, so please keep it. I also like the concrete suggestions made here (starting from line 273), but are they in a right place? Thinking that this bit would belong under the title in section 7, I make some suggestions below.
- See my comment on line 142. Is adaptive heritage something that falls into a wider category of adaptive management? Is it more specific or does it encompass totally new elements compared to adaptive management?
- This is more interesting: trying to concretize how does this new management methodology look like. I long for real-life examples already… The same feeling goes with the wider landscape management “method” – maybe reconsider showing examples earlier? Like in lines 256-262 you so nicely do.
- Again, I feel repetition. Please check and omit parts that repeat this “static versus dynamic” issue. It is not a bad outcome if the text shortens in the process.
- These constraints are not very striking to me, more so they seem practical (like that there is a limit to the spatial and temporal scales of operation). Maybe elaborate or clarify what is the striking weakness here, or choose somewhat milder expression.
- How does adaptive heritage differ from the term adaptive management? Does it have something to do about his spatial scale issue, or is there another reason to use a new term?
- There seems to be something (maybe a verb) missing from this sentence.
- Very well put, here is the main challenge in a nutshell. I am not sure if this bit can be presented a bit further from the lines 115-125, but if it could, I would love to see the lines 115-125 maybe at the very beginning of the part 2, setting the scene as it is, and then moving to compare Western and African/Asian ways to see the change, and maybe finally coming back to this (most important) bit, pointing the need to change our viewpoint on PA management all over the world. So the contents of the text is mostly fine, but I would have another look at the storyline in parts 1 (intro) and 2, and remove repetitions to make it even smoother. You could also think about if part 2 is actually needed or should it be fused to be part of introduction?
- See my comment on line 47 above. I feel that this (nice) paragraph could be presented earlier in the text, more like an introduction or presenting the status quo. This gives credit to the PA managers that they are not dummies that just want to store nature in a museum (to put very bluntly, this was the feeling I got from the intro). What is new, is the RATE at which things are changing now and in the near future. That is the big challenge we need to face and we need new tools to cope with. See the next comment immediately below.
- I would put the reference 17 within the same parentheses with “and lost”. Same elsewhere in the text (but maybe the paper has a different instruction?).
- Nothing wrong in this paragraph, except that I have a faint feeling that I have read this content before. Please check if this is repetitive e.g. with lines 42-49 or partly with 57-71. I think this paragraph is well written, so maybe consider keeping this one and omitting the text before where the same was said more bluntly.
- “Managers need a better understanding…” This is a very good sentence, right on the point!
- If a reader does not have profound knowledge on World Heritage sites or OUVs, they are left to wonder about the word “value” here. Please elaborate and/or give some common examples here (or, if they are given soon in the text below, point to this). Are we talking mostly about ecological values? Aesthetics? Or both, or something else? This is especially important as you go on about how these values are “in the eye of the expert” and how they tend to get “fixed” in a process of setting up the heritage site.
- Degradation is change. NEGATIVE change! PAs usually are designated in a way that allows for evolutionary changes (in fact, this is one of the basic reasons to protect a site, to allow the populations within to breed, spread and finally evolve). So not all change is considered inherently bad when designating a PA. The sentence as it stands now is too simplifying, please rephrase.
- I am used to think that biodiversity underlies ecosystem services, and thus I would rephrase the sentence: “These benefits include biodiversity and ecosystem services such as aesthetics…”
- I would love to have a clearer understanding if these statements that you make in the titles are something that you criticize or stand for (now there is both, compare titles 2 and 3). You could state this at the very end of the introduction, giving a reading instruction in the same time. E.g.: “In the next chapters (2-5), we will go through typical viewpoints on natural heritage management that are, in our opinion, somewhat or totally outdated. We then present fresh ideas how to adaptively manage natural heritage in chapters 6 and 7. In chapters 8 and 9 we highlight innovative examples of adaptive heritage, and we conclude our findings in chapter 10.” There is something like this in lines 143-147, but I would like to see it earlier in the text, given that part 2 is still going to be there (and not fused into introduction).
- As ”adaptive heritage” is a totally new term (at least for me) it is important to add to the keywords “adaptive management” which is more common.